# Targeting DNA Repair, Cell Cycle, and Tumor Microenvironment in B Cell Lymphoma

**DOI:** 10.3390/cells9102287

**Published:** 2020-10-14

**Authors:** Paul J. Bröckelmann, Mathilde R. W. de Jong, Ron D. Jachimowicz

**Affiliations:** 1Max Planck Research Group Mechanisms of DNA Repair, Max Planck Institute for Biology of Ageing, 50931 Cologne, Germany; Paul.Broeckelmann@age.mpg.de; 2Department I of Internal Medicine, Center for Integrated Oncology Aachen Bonn Cologne Duesseldorf (CIO ABCD), University of Cologne, 50937 Cologne, Germany; 3Department of Hematology, University Medical Center Groningen, University of Groningen, 9713 GZ Groningen, The Netherlands; m.r.w.de.jong@umcg.nl; 4Department of Pathology and Medical Biology, University Medical Center Groningen, University of Groningen, 9713 GZ Groningen, The Netherlands; 5Cologne Excellence Cluster on Cellular Stress Response in Aging-Associated Diseases (CECAD), University of Cologne, 50931 Cologne, Germany; 6Center for Molecular Medicine Cologne, University of Cologne, 50931 Cologne, Germany

**Keywords:** DNA damage response, DSB repair, ATM, B cell development, mantle cell lymphoma, cell cycle, cyclin D1, tumor microenvironment, STING, immunotherapy

## Abstract

The DNA double-strand break (DSB) is the most cytotoxic lesion and compromises genome stability. In an attempt to efficiently repair DSBs, cells activate ATM kinase, which orchestrates the DNA damage response (DDR) by activating cell cycle checkpoints and initiating DSB repair pathways. In physiological B cell development, however, programmed DSBs are generated as intermediates for effective immune responses and the maintenance of genomic integrity. Disturbances of these pathways are at the heart of B cell lymphomagenesis. Here, we review the role of DNA repair and cell cycle control on B cell development and lymphomagenesis. In addition, we highlight the intricate relationship between the DDR and the tumor microenvironment (TME). Lastly, we provide a clinical perspective by highlighting treatment possibilities of defective DDR signaling and the TME in mantle cell lymphoma, which serves as a blueprint for B cell lymphomas.

## 1. Introduction

DNA damage activates a complex signaling network, commonly referred to as the DNA damage response (DDR) [1]. The DDR induces cell cycle checkpoints and therefore allows time for DNA repair, or—if the lesions are beyond repair capacity—leads to the activation of cell death pathways [1]. These checkpoints are the G1/S checkpoint, the S phase checkpoint, and the G2/M checkpoint. Once DNA damage occurs, DDR proteins will accumulate at damaged sites to stabilize broken ends, attract repair proteins, or even facilitate in the repair. Mammalian cells evolved a series of distinct DNA repair mechanisms to remove structurally distinct DNA aberrations throughout the different phases of the cell cycle [2]. These pathways include mismatch repair (MMR), base excision repair (BER), trans-lesion synthesis (TLS), the Fanconi anemia (FA) pathway, and two distinct dominant DNA double-strand break (DSB) repair pathways, namely, the error-prone classical non-homologous end joining (cNHEJ) pathway, and the error-free homologous recombination repair (HR) pathway [2,3,4,5]. Error-prone cNHEJ operates throughout the cell cycle and directly ligates the processed DSB ends, whereas error-free HR is restricted to late S- and G_2_, when a repair template is available [2]. HR involves the binding of the MRE11-RAD50-NBS1 (MRN) complex to tether DSB ends and to recruit and activate the proximal DDR kinase ataxia telangiectasia-mutated (ATM) [6]. ATM phosphorylates a plethora of substrates, such as CHK2, p53, UBQLN4, MDC1, and MRN complex components [6,7]. In particular, ATM-dependent phosphorylation of NBS1, RAD50 and MRE11, and downstream components of DNA end resection, such as CtIP contributes to the activation of HR and may also create a positive feedback loop that maintains ATM activity [6,8]. The overall performance of DNA repair pathways, such as the right choice between these pathways is essential for correct DNA repair [2]. At the same time, activation of the cell cycle checkpoints will set pathways in motion to halt cell cycle progression. An important example of a protein that bridges the gap between the DDR and the cell cycle is the WEE1 kinase. WEE1 phosphorylation of CDK1 and CDK2 keeps it in an inactive state, thereby preventing cell cycle progression [9]. Upon DNA damage, activation of the checkpoint kinase 1 (CHK1) by the DDR results in the activation of WEE1, halting the cell cycle and either allowing repair or inducing apoptosis [10].

Components of DNA repair signaling and the cell cycle machinery are among the most frequently mutated genes in human malignancies, including B cell lymphomas [2,11,12,13]. The fundamental process of B cell development is crucially dependent on functional DNA repair and cell cycle control mechanisms and thus deficiencies within these pathways do not only promote lymphomagenesis but are also frequently found in B cell lymphomas [14]. For example, ATM is recurrently mutated in up to 50% of mantle cell lymphoma (MCL) [15]. As a functional consequence, defects in ATM have been associated with resistance to conventional chemotherapy treatment in cancer [16]. In addition, Cyclin D1, a D-type cyclin which associates with CDK4 or CDK6 to control the progression to the G1 phase and the G1/S phase transition, is overexpressed typically due to a t(11;14) (q13;q32) translocation in the majority of MCL patients, which leads to uncontrolled cell proliferation [17]. While 1st-line therapy with intensive chemo-immunotherapy induces disease remissions, relapse is inevitable.

In this review, we will focus on the role of rewired DDR signaling and cell cycle control in B cell lymphomagenesis, and elucidate the therapeutic potential that these alterations reveal. We will place a specific emphasis on the intricate relationship between the cell cycle, the DDR, and the tumor microenvironment (TME). The TME is heterogeneous across cancer entities and composed largely of immune and stromal cells. It provides a protective niche for tumor cell proliferation and facilitates immune evasion, ultimately resulting in cancer growth and therapeutic resistance. Leveraging the TME regained attention with the advent of immune checkpoint blockade (ICB), which reverses inhibitory signaling and the anergic state of immune effector cells in the TME. An accumulating body of evidence implicates a direct link between DDR defects and responsiveness to ICB [18,19]. For instance, microsatellite instability/defective mismatch repair (MSI/dMMR) has been approved by the FDA as a pan-cancer DDR biomarker for predicting response to ICB [20]. An impaired DDR is also associated with decreased antitumor immunity [21]. Increasing evidence suggests that targeting cell cycle dysregulation has the potential to increase the efficacy of ICB in otherwise unresponsive tumors [22,23]. We review the existing evidence to develop novel therapeutic approaches by harnessing DDR deficiency and leveraging the increasingly understood TME features using MCL as a blueprint for B cell lymphomas.

## 2. Role of DNA Repair in B Cell Development and Lymphomagenesis

Initial B cell development takes place in the bone marrow in the absence of antigen. It is a highly regulated process in which DDR signaling plays an important role (Figure 1). More specifically, developing B cells undergo programmed DNA DSBs at precise loci within their genome as they assemble antigen receptor genes required for developmental processes [24]. B cell antigen receptors (BCRs) are composed of immunoglobulin heavy (IgH) and light (IgL) chains encoded by different genes, which are assembled from the recombination of variable (V), diversity (D), and joining (J) gene segments [24]. V(D)J recombination initiates when the recombination-activating gene 1 (RAG1) and RAG2 endonuclease cleaves DNA at the border of two *Ig* gene segments and their flanking recombination signal sequences in G1 phase B cells [25]. RAG DSBs are then repaired exclusively by error-prone cNHEJ to assemble a complete *Ig* gene, enabling Ig chain expression on the surface of maturing B cells which typically harbor insertion-deletion mutations (indels) [26]. This process leads to the activation of ATM, which in a positive feedback loop maintains the stability of RAG induced DSBs [27]. In response to BCR activation, mature B cells then again undergo developmentally programmed DSBs within the constant region of the *IgH* locus in a process termed class switch recombination (CSR) [28]. DNA repair deficiency during CSR leads to impaired immunoglobulin switching and to chromosomal translocations involving the *IgH* locus. To illustrate, B cells deficient for ATM display impaired CSR and frequently generate chromosomal translocations involving the IgH chain locus [29]. Similarly, in Atm^−/−^ mice, CSR has been shown to be reduced by 75% [30]. Clinical and in vivo studies reveal a similar pattern: patients with inherited dysfunctional ATM alterations (termed Ataxia-telangiectasia, A-T) and mice lacking Atm function are immunocompromised and have an increased risk of developing lymphomas [31].

Mature non-neoplastic B cells with successful V(D)J recombination and that express functional BCRs leave the bone marrow and migrate to peripheral lymphoid tissues. Upon exposure to an antigen in the secondary lymphoid organs, these cells enter the primary follicle to form the germinal centers (GCs) that comprise two different regions which are termed the dark zone (DZ) and light zone (LZ). GCs are specialized microenvironments and are critical for the formation of long-lived plasma cells and memory B cells. Within these structures, B cells undergo somatic hypermutation (SHM) and clonal selection based on the affinity of the BCR for the immunizing antigen [32]. Recent evidence suggests that CSR ceases upon the onset of SHM and—unlike SHM—mainly takes place before mature B cells differentiate in GCs [33]. SHM induced lesions are repaired through the DNA repair pathways MMR and BER, both of which are unaffected by ATM deletion [34]. The process of SHM allows the determination of whether a B cell has experienced the GC and provides some evidence for the cell-of-origin in situations in which faulty B cell development has led to B cell lymphoma. The intrinsic force of B cell development—the sequence of V(D)J recombination, CSR, and SHM—can lead to somatic mutations within numerous oncogenes that are associated with lymphomagenesis and their cell-of-origin stemming from the pre-GC, GC, or the post-GC [35,36]. The GC B cell, for instance, is at high risk of undergoing malignant transformation due to genetic lesions that occur in GC-specific DNA remodeling events for Ig affinity maturation [37]. Beyond mere B cell development, several alterations in DNA repair pathways have been linked to lymphoma susceptibility [38]. This is highlighted by human genome instability syndromes like A-T or Nijmegen-breakage syndrome (alterations in *NBS1*) which typically present with lymphomas at an early age [39]. Descriptive and functional genomic studies provide further evidence for the impact of DNA repair genes on lymphoma risk [40,41,42]. Despite its importance, the mechanism behind DDR alterations and lymphomagenesis remains unclear. Recently, *Atm* reactivation in initially *Atm*-deficient B cell lymphomas has been shown to induce tumor regression strongly corroborating the suppressive function of intact DDR on lymphomagenesis [43]. Thus, addressing the rewired DDR response for B cell lymphoma offers intriguing possibilities for genotype-stratified treatments.

## 3. Role of the Cell Cycle in B Cell Development and Lymphomagenesis

Cell division during B cell development is a strictly controlled process, which only occurs at specific moments during development or during an immune response after encountering antigens. DNA replication during proliferation is associated with HR, whereas VDJ rearrangement is associated with cNHEJ, making these two events mutually exclusive. Early B cell development is thus characterized by waves of V(D)J recombination followed by waves of proliferation. This separation ensures that no events of recombination will occur during DNA replication, thereby preventing an increase in the mutation rate. Expression of RAG2 is therefore mostly restricted to the G0 and G1 phase of the cell cycle and quickly degraded upon entering the S phase [44]. This degradation of RAG2 is initiated by the cyclin A/CDK2 complex, which promotes ubiquitination and degradation of RAG2 by Skp2-SCF [45]. Uncoupling this cell cycle regulation from RAG2 expression has been shown to promote aberrant recombination, and combined with p53-deficiency leads to genomic instability and lymphomagenesis [46]. A similar regulatory mechanism is in place for activation-induced cytidine deaminase (AID): the expression of nuclear AID and AID-induced DSBs are observed almost exclusively in the G1 phase of the cell cycle [47] and prolonged-expression of AID enhances DNA damage at off-target sites [48].

As one of the hallmarks of cancer, genetic alterations to cell cycle genes are found in most B cell lymphomas. The most commonly found genetic alteration is of the tumor suppressor gene *TP53*, with mutations found in MCL (30%), Burkitt lymphoma (30%), diffuse large B cell lymphoma (DLBCL) (8%), and follicular lymphoma (FL) (6%) [49], resulting in a loss of cell cycle arrest. A similar effect is induced by mutations in *CDKN2A*, which codes for tumor suppressor proteins p16 and p14, resulting in the simultaneous inactivation of retinoblastoma (Rb) through the loss of CDK4/6 inhibition by p16, and loss of p53 activation by p14. Deletions of *CDKN2A* are observed in 30% of MCL cases and combined with *TP53* aberrations are associated with chemoresistance in MCL [50]. Adjacent to *CDKN2A* on chromosome 9 are *CDKN2B* and *CDKN2C*, which code for p15 and p18, respectively, and similarly inhibit CDK4/6 to prevent G1 phase cell cycle progression. Similarly, a loss of Rb protein expression (12% in DLBCL and 12% in FL) eliminates its suppressor effect in the G1/S-phase transition. Taken together, these aberrations are focused on enhancing cell cycle progression through the G1 phase, driving proliferation. Moreover, mutation of oncogenes with cell cycle regulator function such as BCL6 (30% in DLBCL) and MYC (70% in Burkitt lymphoma) also drive high proliferation levels of lymphoma cells [51]. Finally, many lymphomas harbor genetic aberrations in the *AURKA* and/or *AURKB* genes (40% in DLBCL and 80% in Burkitt lymphoma), which code for the mitotic aurora A and aurora B kinases. Aurora A is essential for proper formation and stabilization of the microtubule during mitosis, and overexpression has been shown to induce aneuploidy and spontaneous growth of lymphomas, Aurora B is essential for proper chromosome alignment and accurate chromosome segregation [52]. Taken together, these data present MCL, and B cell lymphomas in general, as a classical example for the most important tool of malignant transformation: bending and harnessing the power of the cell cycle to enable unlimited replication. Especially in B cell development, where cell division is more tightly controlled in order to prevent aberrant genetic alterations, mutations in proliferation genes have a much greater impact, making lymphomas the ideal candidate for cell cycle inhibitor therapies.

## 4. Harnessing DDR- and Cell Cycle Defects for the Treatment of B Cell Lymphoma

Based on the extensive knowledge on the cell cycle and genetic aberrations of cell cycle-related genes in B cell lymphomas, we are now in a position to explore treatment options that directly or indirectly target the cell cycle, of which the most recent and/or common treatments are summarized in Figure 2. As of today, many clinical trials are underway for CDK4/6 inhibitors, such as palbociclib/PD-0332991 (phase 2 in MCL), abemaciclib/LY2835219 (phase 2 in refractory/relapsed MCL), ribociclib/LEE011 (phase 1 in lymphomas), and voruciclib (phase 1 in relapsed/refractory MCL, B cell malignancies and AML). Promising results were also seen for palbociclib combined with BTK inhibitor ibrutinib [53] and proteasome inhibitor bortezomib in MCL [54], and combined with DNA damage-inducing treatments such as irradiation [55] or WEE1 inhibitor AZD1775 [56] in other cancers. Further investigation showed that CDK4/6 inhibition has the ability to shift the DDR response from HR to cNHEJ [57], in addition to reducing PARP1 expression levels [58], making it a potential candidate to use alongside DNA repair inhibitors or cancers with DNA damage deficiencies. Inhibitors that target CDK1/2 currently under investigation include dinaciclib/SCH727965 (phase 1 in MCL and non-Hodgkin lymphomas, NHL) and AG-024322 (phase 1 in NHL), in addition to specific CDK2 inhibitors SNS-032/BMS-387032 (phase 1 MCL and advanced B-lymphoid malignancies) and CYC065 (phase 1 in refractory/relapsed CLL). Besides specific CDK inhibitors, pan-CDK inhibitors such as alvocidib/flavopiridol (phase 1/2 in MCL and DLBCL), P276-00 (phase 2 in refractory/relapsed MCL), and AT7519M (phase 2 in refractory/relapsed MCL) are under investigation that target multiple CDKs at the same time, including CDK7 (part of the Cdk-activating kinase (CAK) complex) and CDK9 (part of the multiprotein elongation factor complex for RNA polymerase II). Pre-clinical experiments with CDK1/2 inhibitor seliciclib/roscovitine revealed that treatment of MCL cells resulted in cell cycle arrest at the G2/M phase followed by apoptosis, in addition to causing downregulation of cyclin D1 and MCL-1 protein levels [59]. Similar effects were induced in MCL by alvocidib [60], which also showed efficacy combined with bortezomib in MCL [61]. Moreover, alvocidib has been shown to enhance the radiosensitivity of ovarian carcinoma cells [62], likely caused by its ability to bind to DNA [63], making it another interesting compound for combination with DNA repair inhibitors. Other treatment options that target the cell cycle include inhibitors for aurora kinase A/B, with clinical trials on-going for alisertib/MLN8237 (phase 1 in MCL and NHL), AT9283 (phase 1 in NHL) and chiauranib (phase 1 in NHL), inhibitors for WEE1 with adavosertib/AZD1775 (phase 2 in NHL), and inhibitors for kinesin spindle protein (KSP) with ispinesib/SB-715992 (phase 1 in NHL). Research in other cancers has shown that treatment of ovarian carcinoma cells with alisertib decreased the expression of PARP and BRCA1/2 and enhanced DNA-PKcs activity, stimulating cNHEJ over HR [64] and showed synergism with CHK1 inhibitor rabusertib/LY2603618 [65]. In addition, alisertib showed promising results combined with docetaxel in MCL [66], further expanding its therapeutic potential. The observation that many cell cycle inhibitors alter the DDR, either by tilting the balance towards cNHEJ over HR or by directly influencing DDR protein expression levels, gives us novel therapeutic options. These may thus focus on the simultaneous targeting of both the DRR and cell cycle, especially in B cell lymphomas with additional DDR deficiencies.

## 5. Exploiting the Tumor Microenvironment for the Treatment of B Cell Lymphoma

B cell lymphomas usually arise in a TME reminiscent of lymphoid tissue which is heterogeneous and shaped by the genetic alterations and phenotype of the malignant cells. The TME of B cell malignancies is composed of variable types and numbers of immune cells, stromal cells, extracellular components, and blood vessels. Spatial arrangement and cross-talk between malignant and bystander cells and the host inflammatory response affect these protective niches [67]. Tumor cells are able to orchestrate the TME in their favor to evade an antitumor immune response or ICB treatment e.g., by downregulation of antigen-presenting features, upregulation of inhibitory proteins, or cytokine release [68]. The interaction between the immune infiltrate and malignant cells thereby shapes the TME, and as a result, the potential therapeutic implications of the interplay between DDR deficiencies and TME composition are increasingly revealed [19].

For example, recent preclinical studies suggest that targeting defective DNA repair may act synergistically with ICB, since the cell cycle dysregulation observed limits antitumor immunity: Mutations in *TP53*—found in up to 26% of MCL patients [69]—were associated with less immune cytolytic activity in one pan-cancer study [70]. An altered p53 pathway additionally results in recruitment of macrophages and CD4+ T cells while decreasing CD8+ T cell levels via dysregulated NF-κB or JAK-STAT signaling [71,72]. In a Kras and p53 driven pancreatic mouse model, signatures of cell cycle dysregulation (E2F, G2/M, DNA repair) correlated with low tumor T cell content and immunologically “cold” tumors, resulting in inferior efficacy of anti-PD1 blockade compared to T cell rich “hot” tumors [73]. Similarly, deletions or mutations of *CDKN2A* are more frequently observed in “cold” pancreatic cancers with a non-T cell-inflamed TME associated with reduced cytolytic activity [74]. Interestingly, *CDKN2A* mutations are also observed in 25% of MCL patients and result in cell cycle dysregulation due to increased CDK4/6 activity by lack of endogenous inhibition by the *CDKN2A/B* gene product p16 [75]. Cyclin D1 overexpression resulting in CDK4/6 activation is a hallmark of MCL, and recently, whole genome sequencing of different human tumors revealed a strong link between cyclin D1 amplification and ICB resistance [21]. Clinically, this aligns well with the disappointing response rates observed with single-agent anti-PD1 antibodies in relapsed or refractory MCL patients [76].

This feature of cyclin D1 overexpression is reversed by CDK4/6 inhibition, which results in increased antitumor immunity due to enhanced CD8+ T cell infiltration and activity with increased IL-2 secretion in lung cancer mouse models [77]. An improved antitumor immune response was also observed with the CDK4/6 inhibitor abemaciclib in a transgenic breast cancer mouse model [78]. By suppressing DNMT1 expression, abemaciclib leads to the expression of endogenous retroviral elements by the malignant cells and ultimately resulted in increased ICB responsiveness due to higher interferon and MHC-I signaling [22]. Similar observations were made in syngeneic mouse models and samples derived from melanoma patients [78,79]. Interestingly, CDK4/6 inhibition induces senescence of malignant cells activating senescence-associated secretory phenotypes (SASP) induced by NF-κB and STAT3 signaling, consecutively resulting in increased antitumor immunity [23,78]. In contrast, SASP induces recruitment and activation of immune effector cells such as NK cells, macrophages, and CD8+ T cells limiting tumor growth in tumors with *p53* WT, activated MYC, active Wnt, and low Notch signaling. Such a SASP pattern is observed in the context of CDK4/6 inhibition in *KRAS*-mutant lung and pancreatic cancer models, leading to an effective antitumor immune response [23]. Importantly, alternative SASP may facilitate tumor growth by inducing a state of chronic inflammation establishing an immunosuppressive TME in tumors with *p53* mutations or activated Notch signaling. Targeting the cell cycle machinery beyond CDK4/6, a recent study found enhanced ICB sensitivity due to increased CD4+ and CD8+ T cell immunity with CDK7 inhibition in small cell lung cancer models [80]. While these studies investigated solid tumor models, they provide a mechanistic rationale to investigate ICB combinations with CDK inhibitors in DDR deficient B cell lymphomas such as MCL.

Stimulator of interferon genes (STING) signaling is triggered by cytosolic DNA and is crucial for antitumor immunity e.g., by inducing type 1 interferon production, inducing cell death, and releasing tumor antigens or increased priming and activation of effector T cells [81]. B cell malignancy mouse models show that STING activation with the agonist 3′3′-cGAMP directly induces apoptosis and tumor regression of malignant B cells but not in solid tumors in an interferon independent fashion [82]. Inducing DNA damage in in vivo models of small cell lung cancer via PARP inhibition (PARPi) and/or CHK1 inhibition not only potentiate the efficacy of anti-PDL1 treatment by augmenting cytotoxic T cell infiltration but additionally activate the STING pathway. Notably, knockout of *cGAS* and *STING* reversed the observed synergies of ICB and DDR inhibition [83]. Further supporting activation of STING innate immunity by DDR inhibition, treatment of Brca1-deficient ovarian cancers with PARPi in mice resulted in STING activation, and an improved anti-tumor effect when combined with anti-PD1 ICB [84]. Additionally, DNA damage induced by targeting telomerase sensed by dendritic cells results in STING activation and overcomes ICB resistance in advanced tumor models [85]. Recently, small-molecule STING agonists with potent antitumor activity in various animal models synergizing with ICB were developed, providing promising drugs to further develop STING-based treatments in tumors harboring DDR deficiency [86,87].

As summarized previously, combinations of PARPi and anti-PD1 ICB leads to superior tumor control, and a similar effect was observed when combining PARPi with anti-CTLA4 antibodies in a T cell and IFN-γ dependent way [88]. The improved efficacy of PARPi in combination with ICB compared to single agent PARPi may additionally be due to the upregulation of PD-L1 expression observed in breast cancer models [89]. Supporting this evidence, Sato et al. demonstrated that PD-L1 upregulation is dependent on ATM/ATR/CHK1 kinases in response to DSBs [90]. In line with the previously summarized observation with CDK4/6 inhibitors, co-administration of ICB agents hence potentiates the potential of cell cycle or DDR targeting drugs by leveraging and amplifying the increasing antitumor immunity observed during treatment with these agents.

Taken together, these data provide a strong rationale to develop synergistic therapies in B cell lymphomas targeting the dysregulation of cell proliferation and impaired antitumor immunity. Figure 3 conceptualizes the current understanding of potential mechanisms of synergistic effects of cell cycle inhibitors and ICB, utilizing DDR deficient MCL as a blueprint for other B cell lymphomas. Further mechanistic dissection of the interplay between those two intertwined key features of malignancy by elaborate in vitro and in vivo models is crucial to identify targets and drug candidates for early clinical development. Innovative trial designs such as conditional multi-cohort studies may additionally help to ultimately translate these concepts into routine care and improve outcomes for our patients.

## 6. Remarks and Outlook

Defects in DDR pathways expose important vulnerabilities for the treatment of tumors with inhibitors targeting DNA-PKcs, ATR, and PARP1 [7,91,92]. Agents targeting the cell cycle promote antitumor immunity which is potentiated by simultaneous ICB in various in vivo models, providing a rationale to explore potentially synergistic therapies. Alternative error-prone DNA repair pathways such as cNHEJ, which generate potentially targetable dependencies, may bypass the defective DNA repair caused by ATM and/or *TP53* mutations in MCL. For instance, the DNA-PKcs-dependent cNHEJ pathway becomes essential for the cellular survival of doxorubicin-treated B-NHL cells lacking ATM [93]. Identifying backup DNA repair pathways in ATM defective MCL is therefore critical for identifying targeted treatment options for these patients. Destabilization of the cell cycle regulator proteins during crucial events of genetic recombination has been shown to enhance genomic aberrations in B cells [45,47], making it the seemingly ideal strategy for the treatment of B cell lymphomas. Most of these strategies are focused either on (1) prolonging the G1 phase, thereby stimulating aberrant recombination; (2) stalling mitosis, thereby inducing mitotic catastrophe; or (3) abolishment of the cell cycle checkpoints and thereby allowing unscheduled cell cycle progression. However, many (pan-) CDK inhibitors have failed as monotherapy in the past because of a lack of a clear understanding of the biological pathways involved, many that go beyond simple cell division. A weakness of these inhibitors targeting proteins required for the survival of normal cells is a rather small therapeutic window leading to relevant toxicities in patients. The potential success of targeting the cell cycle as a therapeutic strategy will hence likely depend on identifying synergistic pathways and appropriate patient selection. Identification of synergistic drug partners in synthetic lethal screens could alleviate this problem by increasing the therapeutic window and efficacy. For instance, an siRNA screen in *KRAS* mutant tumors identified CDK1 as a synthetic lethal target [94]. In a similar approach utilizing a pharmacogenomic screen in T cell acute lymphoblastic leukemia, Pikman et al. identified a synergistic potential for CDK4/6 inhibitors with glucocorticoids and mTOR inhibition [95]. Leveraging the concept of synthetic lethality beyond siRNA- and pharmacogenomic screens, CRISPR-based screens are further able to identify novel drug combinations [96]. Combining a drug known to be effective in a certain disease with a CRISPR library, these so-called drug anchor screens, may be able to identify previously unmarked context-dependent oncogenes [97]. Applying an in vivo CRISPR screen geared to identify correlates of immune-mediated lethality, extends this approach beyond cell-autonomous targets [98]. A recent study evaluated 2700 candidate genes in anti-PD1 treated B16 syngeneic mouse models, identifying immune evasion genetic contexts such as *JAK1* and *B2M* loss of function [99]. Applied ideally in paired syngeneic mouse models harboring loss of function in the identified gene and a wild-type control, this ultimately facilitates the discovery of drug targets able to counteract immune evasion in the previously identified context-specific immune evasion signature [96]. These concepts, such as immune-stimulating drug combination therapies, are of immediate relevance to future drug discovery and immunotherapy development in B cell lymphomas, such as for MCL, which is largely resistant to single-agent ICB [76]. In conclusion, the complex interplay of defective DNA repair, cell cycle control, and the TME raises intriguing possibilities for targeted treatment approaches in B cell lymphomas that to date represent tremendous challenges and cause high mortality.

## Figures and Tables

**Figure 1 cells-09-02287-f001:**
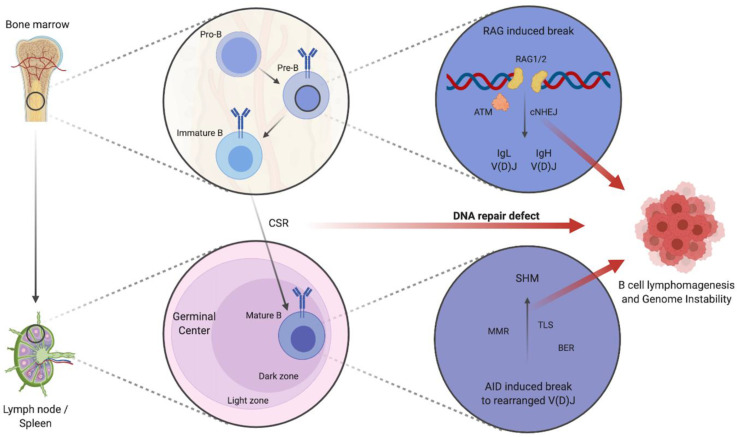
Schematic depiction of B cell development and impaired DNA repair contributing to B cell lymphomagenesis and increased genome instability. Schematic figure depicting the maturation of pro-B cells to immature B cells in the bone marrow by cNHEJ mediated IgL and IgH V(D)J. CSR and consequent SHM of mature B cells in the dark zone of GCs is mediated by further DNA repair mechanisms: e.g., MMR, BER, and TLS. Defects in DNA repair pathways contribute to B cell lymphomagenesis and lead to genome instability with the increased occurrence of translocations, fusions, deletions, or breaks/gaps. Abbreviations: IgL: immunoglobulin light chain, IgH: immunoglobulin heavy chain, V(D)J: variable (V), diversity (D), and joining (J) gene segments, cNHEJ: classical non-homologous end joining, ATM: ataxia telangiectasia mutated, CSR: class switch recombination, SHM: somatic hypermutation, MMR: mismatch repair, BER: base excision repair, TLS: trans-lesion synthesis, AID: activation-induced cytidine deaminase, RAG1/2: recombination-activation gene 1/2. Created with BioRender.com.

**Figure 2 cells-09-02287-f002:**
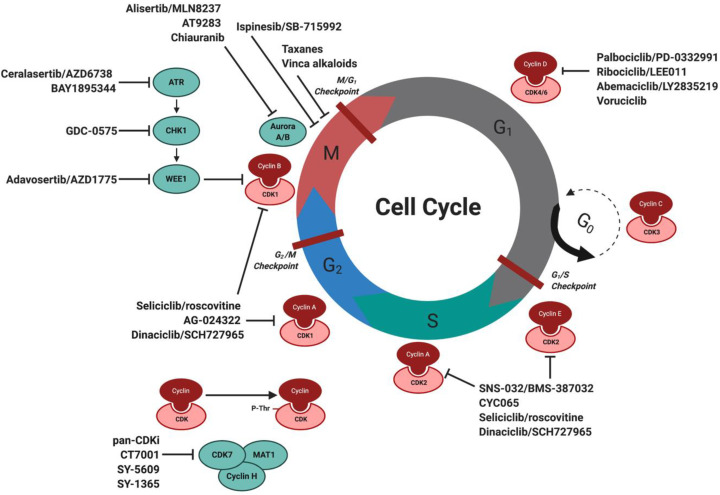
Schematic overview of the cell cycle and cell cycle targeting inhibitors in MCL and other B cell lymphomas. Progression of the cell cycle is facilitated by specific interaction and activation of CDKs with cyclins to allow cell cycle progression. Listed are cell cycle inhibitors currently under investigation for the treatment of MCL and B cell lymphomas, in addition to upstream targeted inhibitors against DDR proteins. Abbreviations: CDK: cyclin-dependent kinases, MCL: mantle cell lymphoma, DDR: DNA damage response. Created with BioRender.com.

**Figure 3 cells-09-02287-f003:**
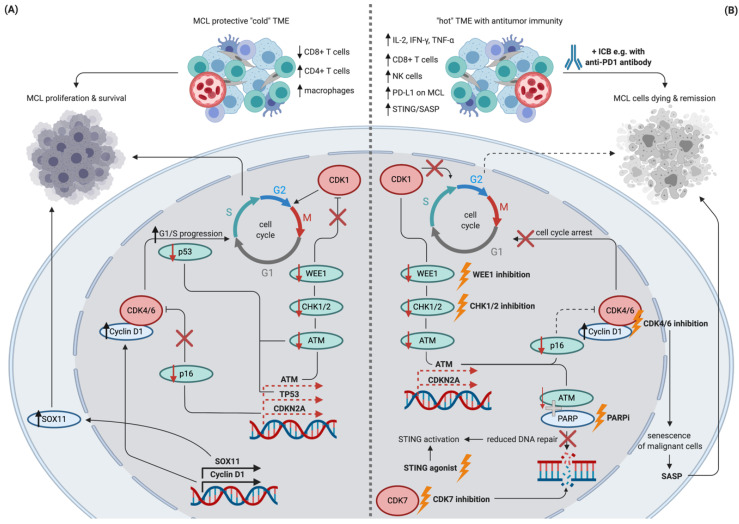
Schematic depiction of potential synergistic effects of DNA repair inhibition, cell cycle inhibition, and immune checkpoint blockade in MCL. Schematic figure showing selected features of treatment naïve MCL resulting in the proliferation of malignant cells (panel **A**) in contrast to potentially synergistic effects of the currently available treatments targeting cell cycle inhibition in combination with ICB (panel **B**). Dashed lines indicate a lower than normal transcription/function. Targetable cell cycle machinery and respective interventions are highlighted in bold and by lightning symbols. Abbreviations: MCL: mantle cell lymphoma, TME: tumor microenvironment, ICB: immune checkpoint blockade, SASP: senescence-associated secretory phenotype, STING: stimulator of interferon genes, PARPi: PARP inhibition. Created with BioRender.com.

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
