# Peer review of "Targeting DNA Repair, Cell Cycle, and Tumor Microenvironment in B Cell Lymphoma"

_cells, 2020, doi:10.3390/cells9102287_

Round 1

Reviewer 1 Report

The manuscript is well written; I found it very interesting because points out the potential relationship between B cell development and cell cycle. 

I would have few minor comments, however:

Title: "targeting deficiencies does not make much sense to me; I would rather write " targeting DNA repair, cell cycle.....

Just a general observation: the author talks about cell cycle in several ways: Sometimes, cell cycle regulation in specific phase ( line 125 on): here they talk about the VDJ and RAG activity which is regulated through the cell cycle phase.

Threafter, from line 165 on, they describe figure 1 where all (or almost all) the chemicals block cell cycle progression or inhibit cell cycle checkpoint.

These different aspects of cell cycle should be discussed a little more at the end of the review.

The authors talk about the tumeor microenvironment (TME) at the line 69 and give a little description about TME from line 204 on.

I guess they should describe what TME is and why it is likely to affect tumor development at the beginning of the review.

Finally, I would add a figure of VDJ recombination depicting how VDJ interact with SHM od MMR to increase genome instability (?)...This could help the readears to undestands why DNA repair is targeted.

After all, I believe that the review is quite appropriate for publication in Cells.

Reviewer 2 Report

Exploiting the DNA damage response (DDR) signaling and cell cycle control pathways for killing cancer cells represents an ongoing target for cancer therapy. In the present review, Bröckelmann et al discuss the role of DDR and cell cycle pathways in B cell development and lymphomagenesis. They also present potential mechanisms for harnessing DDR- and cell cycle defects and exploiting the tumor microenvironment for the treatment of B cell lymphoma. Although this review only briefly touches on the molecular mechanisms of DDR and the cell cycle, it provides a comprehensive summary of chemical compounds targeting DDR and cell cycle components that are employed in clinical trials and highlights the need for the development of synthetic lethality approaches for cancer therapy. I therefore believe it is a well-organized and timely study and support its publication.

Specific comments

  1. Line 156: please change to AURKA and AURKB genes.
  2. Line 157-158: While Aurora A is required for optimal spindle formation as stated by the authors, Aurora B is essential for proper chromosome alignment and accurate chromosome segregation.
  3. Line 192 and Figure 1: Aurora A-inhibited cells exhibit premature exit from mitosis and apoptosis whereas Aurora B-inhibition leads to polyploidy and cell death. It is therefore appropriate to place AuroraA/B inside M and not at the G2/M border in Figure 1.
